# Effects of Heat Reflux on Two-Phase Flow Characteristics in a Capillary of the ADN-Based Thruster

**DOI:** 10.3390/mi13040597

**Published:** 2022-04-10

**Authors:** Zhuan Yan, Xuhui Liu, Yusong Yu, Jie Cao, Xiaodan Liu, Shurui Zhang

**Affiliations:** 1Hydrogen Energy and Space Propulsion Laboratory, School of Mechanical, Electronic and Control Engineering, Beijing Jiaotong University, Beijing 100044, China; 20121394@bjtu.edu.cn (Z.Y.); 18221106@bjtu.edu.cn (X.L.); 21121387@bjtu.edu.cn (S.Z.); 2Beijing Institute of Control Engineering, Beijing 100190, China; xhliu99@163.com; 3China North Engine Research Institute, Tianjin 300400, China; jackcao99@163.com

**Keywords:** ADN-based propellant, thermal reflux, microscale flow, gas-liquid two-phase flow

## Abstract

During the working process of the ADN-based thruster, continuously, heat generated by the chemical reaction in the combustion chamber will transfer along the upstream capillary, the propellant in the capillary continuously absorbs heat under the effect of heat transfer from the wall and undergoes a phase change when the saturation temperature is reached. In this study, effects of the downstream heating temperature (623 K to 923 K) on mass flow rate and pressure change in the capillary were investigated based on the established test platform. Simultaneously, the VOF (volume of fraction) model, and the Lee phase transition model coupled with the Navier–Stokes method was utilized to simulate the spatial distribution of the gas-liquid propellant in the capillary. The results show that the ADN-based propellant firstly formed bubbles on the inner wall surface near the exit of the capillary, and these vapor bubbles moved and grew upstream along the capillary. Due to the cooling effect of the ADN-based propellant inflow, the temperature distribution of the front chamber and capillary gradually reached equilibrium. Bubbles were constantly generated in the capillary, and as the heat reflux intensified, the total volume of bubbles in the capillary continued increasing. Single-phase flow, annular flow, wave flow, and segment plug flow appeared sequentially along the axial direction of the capillary, and the proportion of gas phase volume fraction at the capillary outlet section gradually increased.

## 1. Introduction

With the rapid development of manned spaceflight technology, people’s awareness of environmental protection also increases, and higher requirements are placed on high-performance propulsion systems. For this reason, experts from all over the world have successively developed “green propellants” in the recent years. There are several categories of Liquid-based propellants, such as hydroxylamine nitrate (HAN), ammonium dinitramide (ADN), and Hydrazinium nitroformate, etc. [1]. Among them, the ADN-based liquid propellant is a promising type of green propellant with high energy and low toxicity. For the past few years, it has received extensive attention in the military and aerospace fields, representing the research direction and development trend of aerospace propulsion technology, and has a good prospect of application [2].

In the actual working process, the ADN-based propellant in the tank firstly passes through the catalytic bed through a transport pipeline or a capillary. The ADN-based propellant undergoes an ADN catalytic decomposition reaction on the surface of catalytic particles, producing a large amount of oxidative small molecular intermediates, e.g., NO_2_, N_2_O, NO, and releasing large amounts of heat. After the reactants enter the combustion chamber, the oxidizing substances react with methanol and further release heat. Finally, the high-temperature and high-pressure gas is ejected through the nozzle to generate thrust [3,4].

Under the conditions of small thrust, the engine adopts a capillary with inner diameter of approximately 0.1 mm as the transport channel of the ADN-based propellant. In the process of working, the propellant in the capillary may undergo a phase change due to heat reflux, i.e., downstream heat transfer. When the critical temperature is reached, gas resistance or even capillary explosion occurs [5]. In order to suppress the heat reflux for this microthruster, the European Astrium Company added a copper ring at the junction between the injector and the combustion chamber, which significantly mitigated this problem [6]. The two-phase flow problem was suppressed by the thermal control at the injector by ECAPS and BUSEK of the United States [7]. After adding a heat dissipation structure with a thermal conduction cross-sectional area of 2.5 mm^2^ downstream of the capillary, Liu Xuhui and others from the Beijing Institute of Control found that the heat dissipation structure has a certain effect on the formation of bubbles and the reduction of flow resistance [8]. 

According to a previous study, the flow with a scale of 0.1-mm diameter is generally considered as the category of micro-scale flow [9,10,11]. In terms of experimental research, Wang and Chang [12] studied the heat transfer process of boiling in linear microchannels by an experimental method and observed bubble confinement and elongation. In terms of numerical simulation, Wentan Wang et al. [13] used the Lattice Boltzmann Method to study the bubble formation and floating process in the microchannel. Mosayeb Shams et al. [14] used the VOF model to solve the bubble distribution and motion process in the microchannel. Lin S et al. [15] used Direct Numerical Simulation to study the formation and development process of vesicular flow in microchannels. Faroogh Garoosi [16] found that the numerical simulation of the VOF model can capture the bubble interface and obtain the bubble morphology when studying the bubble distribution inside the microchannel, which is in good agreement with the experiments. Based on the studies listed above [12,13,14,15,16], there are only a few research works on the flow characteristics in the thruster capillary, which limit the understanding of this problem and the solution of practical problems.

In this study, the combination of experiment and simulation was used to study the process of propellant flow heat transfer and phase change in the capillary of ADN-based thrusters under heat reflux for the first time. Based on the established ADN-based propellant capillary flow experimental setup, the heat reflux phenomenon of the 0.2 N ADN-based thruster was reproduced. The influence of heat reflux on two-phase fluid flow in a capillary was also studied by measuring the flow rate and pressure change. The temperature distribution of the capillary was photographed with the help of an infrared camera. Then, the microscopic characteristics of the two-phase flow in the capillary were analyzed by numerical simulation (NS coupled with VOF).

## 2. Experimental System and Error Analysis

### 2.1. Experimental System

The ADN-based propellant capillary flow experimental setup is shown in Figure 1. The test system is composed of three parts: storage tanks and pipelines system, test section bench and data measurement system. Storage tanks and pipeline system include nitrogen cylinder, propellant storage tank, propellant collection tank, and various pipelines connected to it. The test section bench includes support rods, fixtures, an ADN-based thruster, and pipelines. Data measurement systems include a computer, two flow meters, two pressure sensors, a temperature controller, and several temperature sensors.

This test uses high-pressure nitrogen to pressurize the propellant tank. The high-pressure nitrogen passes through the pressure regulator and pushes the ADN-based propellant from the tank into the capillary. This test uses a stable propane flame to heat the capillary outlet area, and uses a K-type armored thermocouple to obtain the heating temperature of the area, and feeds the temperature data back to the temperature controller, so as to achieve accurate temperature control in the heating process. Heat transfers from the capillary exit area to the upstream capillary through the metal structure, creating a so-called heat reflux phenomenon. Flowing through the high temperature capillary is very likely to be caused a phase change for the liquid propellant. The instrument specifications in experimental tests are shown in Table 1.

### 2.2. Measurement Uncertainties Analysis

The method of Holman [17] is used to analyze the uncertainty of the test. For directly measured parameters, the measurement error can be directly obtained from the accuracy of the measuring instrument.
(1)$$\delta R={\left[{\sum_{i=1}^{n}\left(\frac{\partial R}{\partial F_{i}}\delta F_{i}\right)}^{2}\right]}^{1/2}$$

In this formula, δR is the total uncertainty of the calculated parameter *R*, the total uncertainty is affected by a series of measurement parameters of $F_{i}$, $\partial F_{i}$ is the uncertainty, which corresponds to the non-independent variable $F_{i}$. Export variables can be calculated directly using the above equation. The error of direct measurement here includes the error caused by the measurement of pressure, temperature, and flow, as shown in Table 2.

## 3. Thruster Model

### 3.1. Physical Modeling and Simplification

The ADN-based thruster is mainly composed of the capillary, front chamber, thrust chamber (including catalytic bed and combustion chamber), nozzle, and other components. The front chamber connects the capillary and the thrust chamber together, and the heat generated in the combustion chamber is mainly transferred to the capillary through the front chamber. Figure 2a shows the axial temperature distribution of the ADN-based thruster at heat reflux of 923 K at 0.25 s. It can be found that the black region in the front chamber is the dominant region of heat reflux transfer to the capillary. Therefore, in the present numerical simulation, the ADN-based thruster is simplified in order to reduce calculation and the capillary structure is extracted separately. In the present numerical simulation, the ADN-based thruster is simplified. The capillary structure was extracted separately. The capillary was drawn by the 3D-modeling software SOLIDWORKS [18]. The inner diameter of the capillary is 0.15 mm, the outer diameter is 0.60 mm, the horizontal length of the capillary is 15 mm (see Figure 2b).

### 3.2. Grid Division

Grids were generated using ICEM [19]. Since the simplified model is a 3D component with curved segments (see Figure 3), the O-splitting strategy was used for meshing, and mesh encryption of the inner walls of the capillaries. Geometric2 was selected for the node distribution law of the radial capillary, and the rate of change was set to 1.1, and the node distribution law of the axial capillary was selected as BiGeometric. For the front chamber, there is no need to encrypt its mesh, so BiGeometric is chosen for both its axial and radial node distribution laws. Figure 3 shows a schematic diagram of the meshing of the complete capillary flow model.

In this study, six types of grid solution, i.e., 28,487, 38,920, 49,244, 55,512, 64,801, and 78,939 girds, are selected. The bubble volume in the capillary is compared as the basis for grid independence verification. Figure 4 shows the comparison of bubble volume and relative deviations with different grid numbers. As can be seen from the figure, when the number of grids gradually increases, the bubble volume in the capillary gradually stabilizes. The relative deviation is calculated from the result of the grid cell number of 78,939. The relative deviations of grid cell number 55,512 and 64,801 were 0.74% and 0.63%, respectively, and the relative deviations were within 1%. When the number of the grid cells reaches 55,512, the calculated results have little change and have nothing to do with the number of grid cells. Therefore, considering total grid cell number on calculation accuracy, the calculation in the simulation section of this article uses the mesh distribution with a total grid cell number of 55,512 to calculate the results.

## 4. Simulation Calculation Basics

### 4.1. Control Equations

This study will use the NS-VOF model to simulate the flow of ADN-based liquid propellant in the capillary. Continuity equations for the gas and liquid phase are [20]:(2)$$\partial \alpha_{v}\rho_{v}+\nabla \cdot \left(\alpha_{v}\rho_{v}\vec{V}\right)={\dot{m}}_{lv}$$
(3)$$\frac{\partial \alpha_{l}\rho_{l}}{\partial t}+\nabla \cdot \left(\alpha_{l}\rho_{l}\vec{V}\right)=-{\dot{m}}_{lv}$$
where α is the gas phase volume fraction, subscripts v and l represent the gas phase and liquid phase, respectively. Source item ${\dot{m}}_{lv}$ is the boiling phase change rate at which the liquid is heated by the wall.

The momentum and energy conservation equations for gas-liquid two-phase flow are very close to those for single-phase flow. The specific form is as follows:(4)$$\frac{\partial \rho \vec{V}}{\partial t}+\nabla \cdot \left(\rho \vec{V}\vec{V}\right)=-\nabla p+\nabla \left(\mu \left(\nabla \vec{V}+\vec{V^{T}}\right)\right)+\rho \vec{g}+\vec{F}$$
(5)$$\frac{\partial \rho E}{\partial t}+\nabla \cdot \left(\vec{V}\left(\rho E+p\right)\right)=\nabla \left(k_{eff}\nabla T\right)+S_{h}$$

In these equations, p is the fluid pressure, $\vec{F}$ is the surface tension of the gas-liquid interface, and $S_{h}$ is the latent heat of evaporation due to the thermal phase change of the propellant.

The evaporation and condensation of the propellant is simulated using the Lee model [21]:(6)$${\dot{m}}_{lv}=\sum {}_{i=1}^{n}{\dot{m}}_{v,i}$$
(7)$${\dot{m}}_{v,i}=C_{i}\cdot \alpha_{l}\rho_{l}\frac{T_{l}-T_{sat,i}}{T_{sat,i}}$$
where $C_{i}$ is the time factor characterizing phase transition lag.

The gas-liquid surface force model is given as:(8)$$F=\int_{s\left(t\right)}\sigma k^{\prime }{\vec{n}}^{\prime }\delta \left(\vec{x}-{\vec{x}}^{\prime }\right)dS\approx \sigma k\nabla F$$
where $\vec{n}$ is the normal unit vector of gas-liquid interface, and k is the radius of curvature of gas-liquid interface. The relationship between the two is as follows:(9)$$\vec{n}=-\nabla F,k=\nabla \cdot \left(\frac{\vec{n}}{\left|\vec{n}\right|}\right)$$

At the fluid-solid coupling interface, the fluid-solid heat flow conservation should be satisfied [22]:(10)$${K_{\mathrm{cond}}\frac{\partial T}{\partial n}|}_{wf}=q^{conv}=h_{conv}\left(T_{f}-T_{w}\right)$$
where $K_{\mathrm{cond}}$ is thermal conductivity of solid; $q^{conv}$ is the heat exchange volume; $h_{conv}$ is the is the local convective heat transfer coefficient; $T_{f}$ is the fluid temperature; $T_{w}$ is the wall surface temperature.

The energy equation for heat transfer from the solid is [23]:(11)$$\rho_{s}c_{s}\frac{\partial T}{\partial t}=k_{s}\nabla^{2}T+\overset{\cdot }{S_{r}}$$
where *T*, $\rho_{s}$, $c_{s}$ and $k_{s}$ are the temperature, density, specific heat and thermal conductivity of the solid, respectively, and $\overset{\cdot }{S_{r}}$ is the thermal radiation source term.

This study will use the P1 radiation model to consider the radiative heat exchange between the solid and the external space [24]:(12)$$q_{r}=-\Gamma \nabla G$$
(13)$$\Gamma =\frac{1}{3\left(a+\sigma_{s}\right)-C\sigma_{s}}$$
where $q_{r}$ is the radiation flux, a is the absorption coefficient, $\sigma_{s}$ is the scattering coefficient, *G* is the incident radiation, and *C* is the linear-anisotropic phase function coefficient.

The transport equation is shown in the following equation:(14)$$\nabla \cdot \left(\Gamma \nabla G\right)-aG+4an^{2}\sigma T^{4}=S_{G}$$
where N is the refractive index of the medium, σ is the Stefan–Boltzmann constant, and $S_{G}$ is the radiation source term. Combining the above equations, gives Equation (15):(15)$$-\nabla \cdot q_{r}=aG-4an^{2}\sigma T^{4}$$

### 4.2. Model Assumptions and Boundary Conditions

The following are the assumptions for building a numerical simulation model: (1) the gas-liquid phase fluid in the capillary is treated according to dissociation, the viscosity coefficient of each phase fluid is a fixed value. This calculation assumes that the physical properties of the propellant are constant. (2) The VOF model is used to consider the influence of surface tension between fluids in each phase. The shape and aggregation of each phase are determined by the model (see Figure 5).

The physical properties of ADN-based liquid propellant (mass-to-fraction ratio of ADN, H2O and methanol is 63%:26%:11%) are shown in Table 3.

The pressure inlet of the ADN-based propellant (0.5 MPa) is on the left side of the fluid domain. The far right of the fluid domain is the pressure outlet. The ambient pressure of the thruster is set to 0.1 MPa. The material of the capillary is GH3030 with a thermal conductivity of 25.1 W/m·K. The emissivity of the solid material is set to 0.2. The initial temperature of both the solid and fluid domain is 300 K. The connection between the front chamber and the downstream thrust chamber is set as the temperature boundary. The capillary is assumed to be filled with liquid at the initial moment.

Simulation calculations used the CFD software FLUENT, in which we select the pressure base solver for transient calculations, the pressure-speed coupling used the SIMPLE algorithm, and the pressure term adopts the PRESTO algorithm. In the second-order windward format, the calculation time step is set to 1 × 10^−5^ s, which ensures that the Courant number is less than 0.5 during the calculation process.

## 5. Results and Discussion

### 5.1. Experimental Results

In order to study the flow fluctuation, pressure drop, and heat transfer characteristics of the ADN-based propellant in capillaries under different thermal re-immersion temperature conditions, experimental studies were conducted under 6 working conditions. During the working process of the thruster, the heat of the thruster housing is gradually transmitted upstream. In order to verify the accuracy of the calculated model, a comparison of the results of the test and simulation is given in this study. The results show that when the heating temperature is set to 923 K and the temperature is heated to 1.5 s, the outer surface temperature at about 4.3 mm from the capillary outlet was about 420.1 K, which was close to the test result (398.7 K), and the simulation error was about 5.3%.

Before the experiment, the capillary inlet pressure was stabilized at 0.5 MPa by a pressure regulator, and the test heating temperatures were set as 308 K, 623 K, 673 K, 723 K, 823 K, and 923 K; the flow rate and fluid pressure at the capillary inlet of the capillary are monitored by the flow meter and pressure sensor arranged in the test system, and the real-time data are recorded by the software Mthings (Version: V0.2.0.600, Technology Development Co., Ltd., Changnian (Shanghai)) and CatmanEasy (Version: 5.2.2.19, The Federal Republic of Germany, Hottinger Brüel & Kjaer GmbH, Im Tiefen See 45, D-64293 Darmstadt). The pressure obtained is used to calculate the pressure drop. Infrared thermal imager made by the Teledyne FLIR company (Shanghai, China) was used to detect local temperature changes in the injector.

Different from the conventional channel, the starting point of the two-phase flow in the microchannel is characterized by mass flow and pressure fluctuations. Figure 6 shows the flow rate change in the capillary at different heat reflux. Through Figure 5, it is found that when the outlet pressure of the nitrogen cylinder is set to 0.5 MPa, the mass flow rate is basically maintained at about 0.038 g/s. When the flow rate is stable for 10 s and then begins to heat, between 10 s and 20 s, as the heat re-immersion temperature gradually increases, under different heating conditions, the mass flow rate has a small increase. Heated to the specified temperature after 20 s, it is found that as the heating temperature gradually increases, the flow rate begins to fluctuate. When the heating temperature is less than 673 K, due to the relatively low heating temperature, the ADN-based propellant absorbs heat during the flow process, and no phase change occurs at this time, so the flow rate is basically maintained at about 0.038 g/s. When the heating temperature is increased to 723 K, the mass flow rate decreases rapidly after 32 s lapsed, mainly because the flow mode of the propellant in the capillary changes from single-phase flow to two-phase flow, and some ADN-based propellant vaporizes and expands in volume. At the same time, due to the small channel, the mass flow rate decreases rapidly. When the heating temperature was increased to 823 K, the mass flow rate dropped rapidly after 26 s, and the mass flow rate showed dramatic fluctuations; when the heating temperature was 923 K, the mass flow rate dropped rapidly at the same time as the heating temperature was 823 K, but at this time, the mass flow rate in the capillary appeared at the moment of 0, and the flow rate fluctuations in the capillary were more dramatic. When the average mass flow rate in the temperature stabilization region tends to decrease significantly (from 0.038 g/s to 0.013 g/s) as the heating temperature increases, the Vapor block phenomenon during the operation of the ADN-based thruster is becoming more and more obvious. 

Figure 7 shows the change of the upstream pressure of the capillary at different heat refluxes. It was found that when the injector was not heated, the capillary inlet was about 0.500 MPa. When the heating temperature is lower than 723 K, the upstream pressure of the capillary is basically maintained at about 0.500 MPa. When the heating temperature is greater than 723 K, the mass flow rate decreases rapidly at the same time, the upstream pressure of the capillary suddenly increases, which is mainly due to the phase change of the propellant in the capillary, and the resistance of the two-phase flow is often several times or even dozens of times that of the one-way flow; so, the upstream pressure of the capillary suddenly increases, and the pressure increases from 0.500 MPa to about 0.507 MPa. As the heating temperature increases, the upstream pressure fluctuations of the capillary are more intense.

Figure 8 shows an infrared imaging of the injector surface at different thermal re-immersion temperatures. The results show that as the heating temperature increases, the local high temperature area of the injector gradually becomes larger. This illustration shows that as the temperature increases, the temperature is gradually transmitted upstream under the heating downstream of the capillary.

### 5.2. Simulation Results

Through the experiment, it was found that with the gradual increase of the heat reflux, the flow and pressure fluctuations inside the capillary gradually became obvious. For further exploration, the simulation section of this article sets the hot relapse temperature at 923 K (temperature boundary condition). Figure 9 shows the temperature distribution of the front chamber and the outer wall surface of the capillary from 0 to 0.2 s. As can be seen from the figure, when the 923 K heat reflux is set, the heat is gradually transferred upstream under the heat conduction of the front chamber and the capillary. The temperature distribution of the front chamber and the outer wall of the capillary is uniform, the temperature of the outer wall surface of the front chamber is basically maintained at about 890 K at 0.1 s, and the temperature of the outer surface remains basically unchanged with time. At the same time, it is found that before 0~0.1 s, the temperature of the front chamber and the outer wall surface of the capillary changes greatly during the conduction of the heat reflux upstream. It was found that the outer wall temperature at 4.6 mm from the capillary outlet was 441.2 K at 0.1 s and 420.2 K at 0.2 s, and the temperature rose slowly. The reason is that when the conduction continues upstream after 0.1 s, it is cooled by the ADN-based propellant, resulting in a slow change in the temperature distribution of the outer wall surface.

Yuanzheng Lv et al. [25] argued that for flow boiling in microchannels, the generation and growth of bubbles could cause violent disturbances to the flow. Figure 10 shows a contour of temperature distribution when the first bubble is generated in the capillary. It can be found that the temperature downstream of the injector is axisymmetric. By analyzing the cross-sectional temperature distribution cloud map at 1 mm from the capillary outlet, it can be seen that the temperature of the inner wall surface of the capillary is about 400.3 K. The temperature of the outer wall surface of the capillary is about 403.6 K, and the temperature difference is 0.8%. Due to the continuous flow of the ADN-based propellant, it will have a certain cooling effect on the heat transferred downstream, causing the temperature to gradually decrease radially from the outlet.

Figure 11 shows the gas-liquid phase distribution within the capillary. In order to facilitate the description of the growth process in the capillaries, this article takes the moment of bubble generation as the initial moment. Bubbles first form on the inner wall near the capillary outlet, and the nucleated area of the bubbles develops upstream along the capillary axis. Under the heat conduction of the capillary wall, the formation and growth of bubbles in the capillary is caused. At 0.01 s to 0.05 s, the bubbles in the capillary gradually grow upstream along the wall, and no bubble detachment and merger occur at this time. The flow pattern in the capillary is a continuous flow dominated by the liquid phase, and changes from a continuous flow to a circular flow. As the heat is gradually conducted upstream, and at the same time, due to the small inner diameter of the capillary, the bubbles do not completely detach from the inner wall, and the bubble merger phenomenon occurs. The annular flow in the capillary is transformed into a segment plug-annular flow, and it is found that the gas-liquid phase interface produces a wave-like disturbance wave, and a wave-like flow occurs at this time. At 0.15 s, the phenomenon of complete gasification of the liquid phase ADN-based propellant is generated at the capillary outlet.

Figure 12 shows the bubble distribution of the capillary outlet cross-section at different times. The results showed that the bubbles first appeared at the top of the capillary at 0.01 s. The annular flow pattern at the capillary outlet cross-section after 0.05 s is also shown. When the ADN-based propellant continues to absorb heat in the capillary, the bubbles in the capillary gradually increase and merge. It showed an irregular flow trend in the moment of 0.10 s~0.25 s. The volume fraction of bubbles at the capillary outlet cross-section is 26.3%, 34.9%, 63.1%, 84.2%, 76.7%, and 87.8% in ascending order. Since the capillary is directly connected to the downstream combustion chamber, the flow rate of the ADN-based propellant entering the combustion chamber fluctuates, which in turn affects the downstream thruster performance.

Figure 13 shows a plot of the volume change of bubbles in the capillary at different thermal re-immersion temperatures. It is found that with the increase of the heat reflux (from 623 to 923 K), the time of phase change of the ADN-based propellant in the capillary is advanced (from 0.008 s to 0.004 s). In the time interval 0.06~0.08 s, the volume of bubbles in the capillary rises significantly, and the two-phase flow in the capillary is more obvious. The heat reflux increased from 623 to 923 K. The rate of bubble generation was gradually accelerated, and the volume of bubbles in the capillary increased by 138.5%. The reason for this phenomenon is that the higher heat reflux enhances the fluid-solid heat transfer in the capillary, causing the bubble nucleation zone to move upstream of the capillary, resulting in more ADN-based propellant in the capillary undergoing phase change and generating a large number of bubbles. It can also be explained that reducing the thermal re-immersion temperature is one of the measures to inhibit the phase transition of the ADN-based propellant in the capillary.

## 6. Conclusions

In this study, effects of heat reflux on the two-phase flow characteristics in the capillary of ADN-based thruster have been studied by combining experiments with simulation methods, and the main conclusions are as follows:

(1) The heat reflux phenomenon of the 0.2 N ADN-based single-component thruster was reproduced in the experimental method. Experimental studies have shown that due to the conduction of the heat reflux, the mass flow and pressure change of the ADN-based propellant in the capillary fluctuated significantly. With the increase of the downstream temperature (from 623 to 923 K), the mass flow and pressure in the capillary fluctuate in advance, and the vapor block phenomenon in the capillary occurs.

(2) The NS method coupled with VOF and the Lee model enables the simulation of the flow boiling phenomenon in the capillary of the ADN-based thruster. When the heat reflux is conducted from downstream, the temperature of the outer wall surface of the capillary is increased gradually. The cooling effect of the incoming ADN-based propellant causes the capillary temperature gradually steady.

(3) With the downstream heating temperature of 923 K, the vapor bubble formed and moved upstream along the capillary. Unipolar flow, annular flow, wave flow and segment plug flow appear sequentially in the axial direction. As the heating temperature increases, the proportion of gas phase volume fraction at the capillary outlet section is gradually enhanced, and the two-phase flow of gas and liquid becomes more intense.

## Figures and Tables

**Figure 1 micromachines-13-00597-f001:**
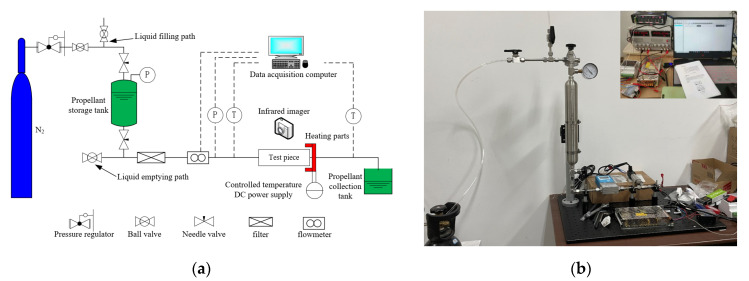
Schematic (**a**) and Image (**b**) of the ADN-based propellant capillary flow experimental system.

**Figure 2 micromachines-13-00597-f002:**
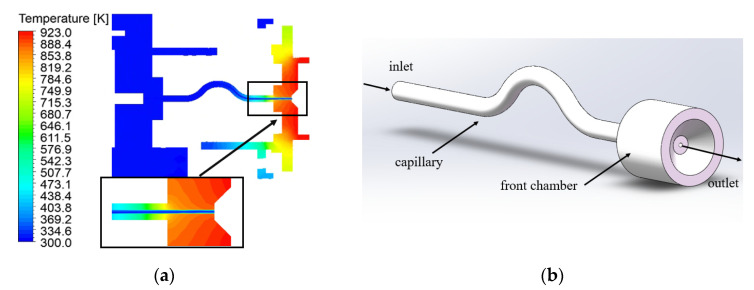
(**a**) ADN-based thruster temperature distributions of the axial plane at 0.25 s; (**b**) schematic diagram of the capillary.

**Figure 3 micromachines-13-00597-f003:**
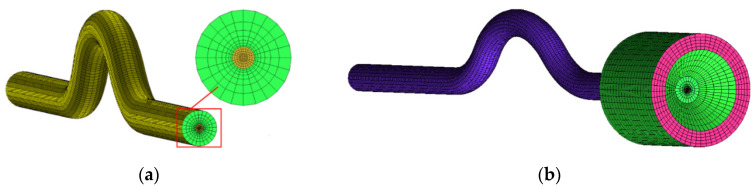
(**a**) Local enlargement of capillary meshing; (**b**) capillary meshing.

**Figure 4 micromachines-13-00597-f004:**
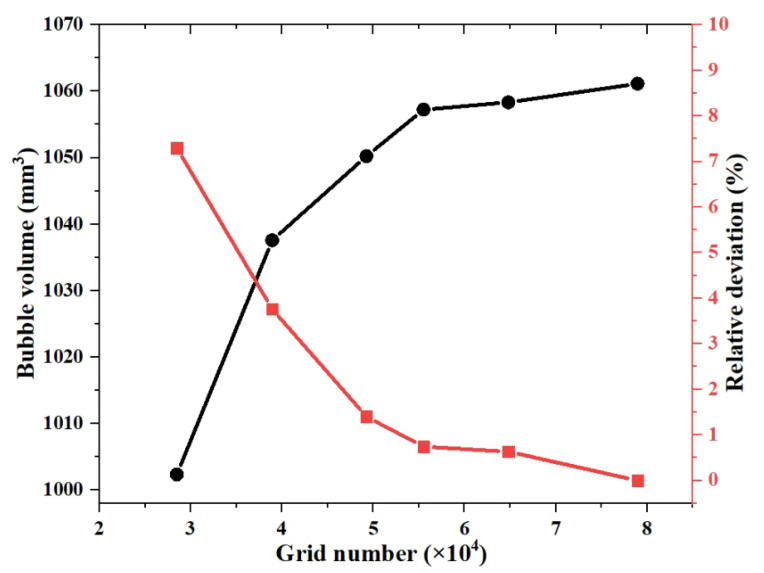
Comparison of bubble volume and relative deviations with different grid cell numbers.

**Figure 5 micromachines-13-00597-f005:**
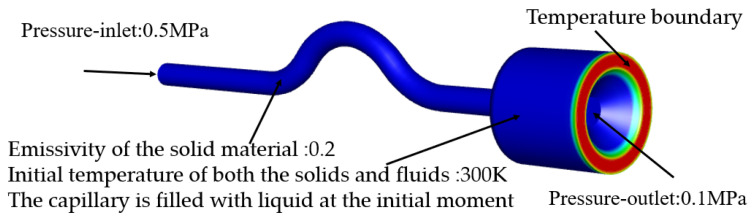
Setting of parameters for capillary flow simulation at the initial moment.

**Figure 6 micromachines-13-00597-f006:**
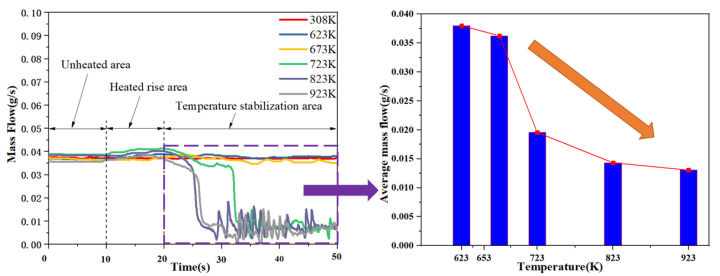
Flow rate change curve in capillaries.

**Figure 7 micromachines-13-00597-f007:**
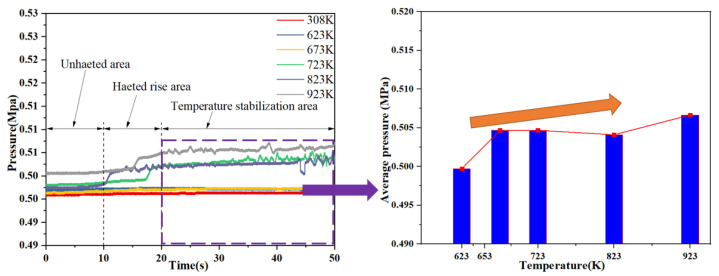
Upstream pressure fluctuation curve of capillary.

**Figure 8 micromachines-13-00597-f008:**
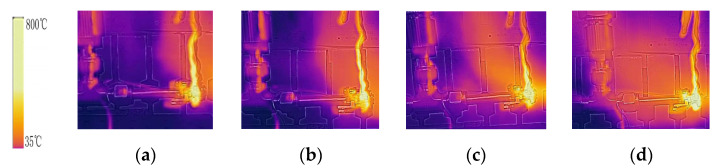
Infrared images of the capillary under different downstream temperatures: (**a**) 623 K (**b**) 723 K (**c**) 823 K (**d**) 923 K.

**Figure 9 micromachines-13-00597-f009:**
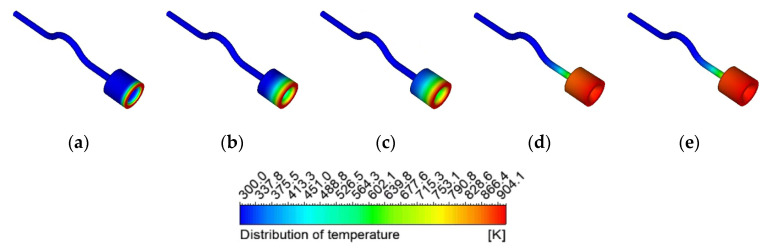
Capillary temperature distribution at different times: (**a**) 0 s (**b**) 0.005 s (**c**) 0.01 s (**d**) 0.1 s (**e**) 0.2 s.

**Figure 10 micromachines-13-00597-f010:**
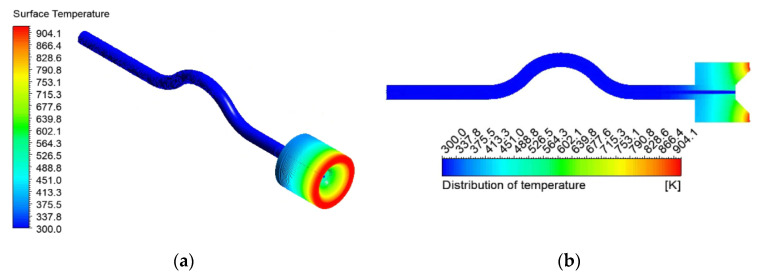
(**a**) Capillary temperature distributions on the outer wall surface at 0.01 s; (**b**) capillary temperature distributions of the axial plane at 0.01 s.

**Figure 11 micromachines-13-00597-f011:**
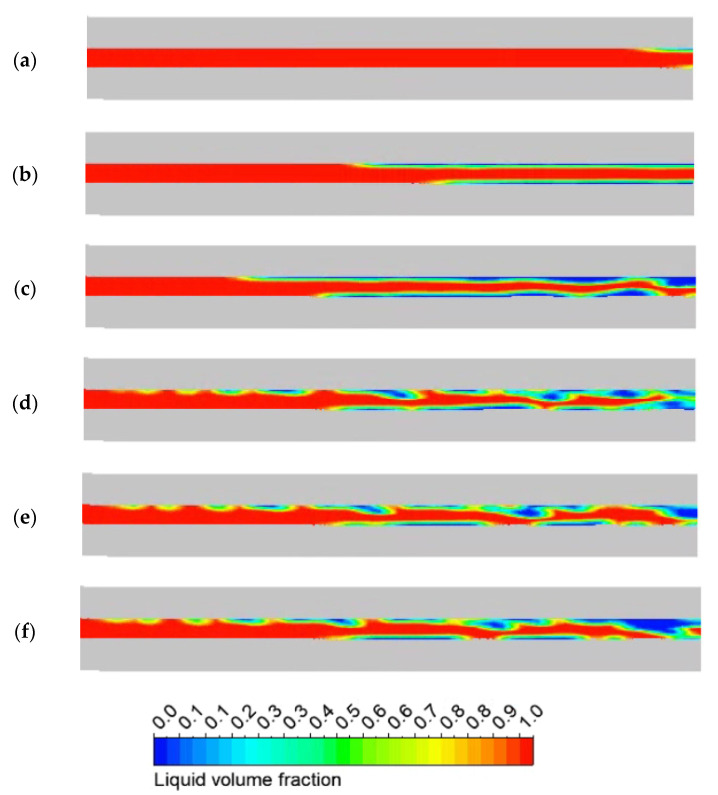
Vapor bubble distributions on axial plane of the capillary at different times: (**a**) 0.01 s (**b**) 0.05 s (**c**) 0.10 s (**d**) 0.15 s (**e**) 0.20 s (**f**) 0.25 s.

**Figure 12 micromachines-13-00597-f012:**
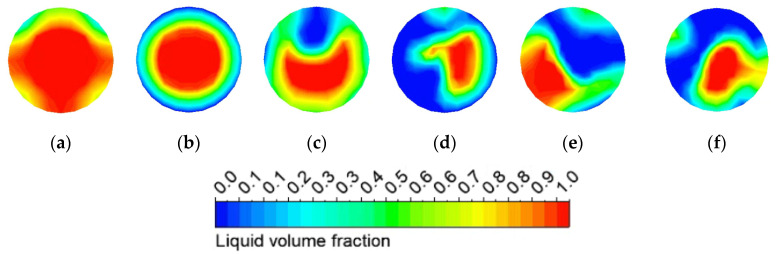
Vapor bubble distributions in capillary outlet section at different times: (**a**) 0.01 s (**b**) 0.05 s (**c**) 0.10 s (**d**) 0.15 s (**e**) 0.20 s (**f**) 0.25 s.

**Figure 13 micromachines-13-00597-f013:**
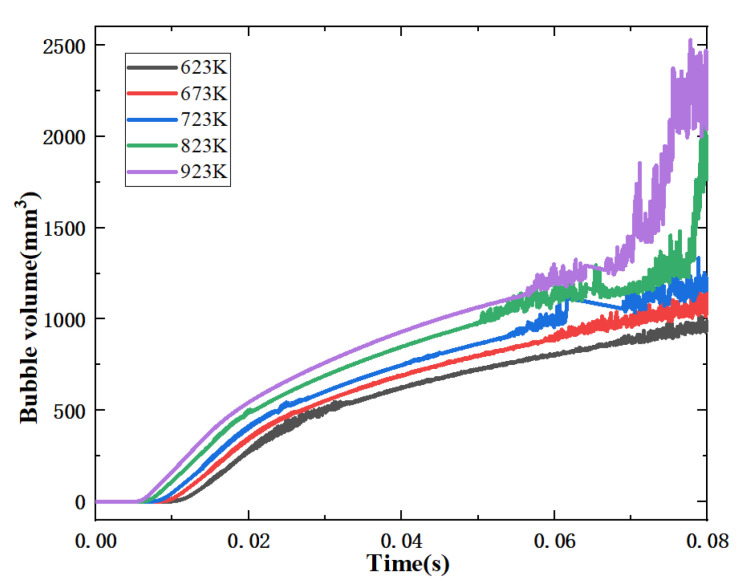
Variation of bubble volume in capillaries at different temperatures.

**Table 1 micromachines-13-00597-t001:** Instrument specifications.

Device Name	Type	The Main Parameters
Mass flowmeter	CX-LFM-4-D-10 mL	Measuring range 0~0.2 g/s work pressure 0~2 MPa
Temperature Sensor	WRNK-162-GH3030	Measuring range 0~800 °C
Cylinder pressure gauge	BYLB-6/25	Measuring range 0~6 MPa
Pressure Sensor	ELE-801	Measuring range 0~4 MPa

**Table 2 micromachines-13-00597-t002:** Uncertainty of direct measurement of test parameters.

Measurement Parameters	Relative Error	Absolute Error
P (Pressure sensor)	0.2%	±0.02 MPa
T (Type K thermocouple)	0.75%	±1.5 °C
L (Mass flowmeter)	0.2%	±0.002 g/s

**Table 3 micromachines-13-00597-t003:** Properties of ADN-based liquid propellants.

Properties	Value
Density (kg/m^3^)	1550
Cp (Specific Heat) (J/kg·k)	2350
Thermal conductivity (W/m·k)	0.8
Viscosity (kg/m·s)	0.0046
Molecular weight (kg/kmol)	124.0562
Boiling point at 0.12 MPa (°C)	80.1
Reference temperature (K)	298.15

## Data Availability

Not applicable.

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
