# Peer review of "Effects of Heat Reflux on Two-Phase Flow Characteristics in a Capillary of the ADN-Based Thruster"

_micromachines, 2022, doi:10.3390/mi13040597_

Round 1

Reviewer 1 Report

This manuscript reports on investigation of effect of hear reflux when the ADN-based thruster is operating, in particular, the associated two-phase flow characteristics. Numerical simulation and experimental works have been conducted in the study. While the existing study has scholarly significance as ADN is a promising ionic green propellant that attracts much interest recently, the manuscript requires a major revamp before it is in publishable form.

Specifically, more in-depth and analysis on the obtained results are expected, along with sufficient explanation.

The following minor revisions are recommended:

  1. L7: ‘… surface near the export of capillary…’, ‘exit’ is the more suitable term than ‘export’.
  2. L32: ‘Liquid based propellants’ instead of ‘liquid unit propellant’
  3. L33: What is ‘hydrazine nitroform’? Is it ‘HNF’? Then, it should be ‘Hydrazinium nitroformate’
  4. L49-50: ‘microthruster’ instead of ‘this type of engine’
  5. L51-52: ‘The two-phase flow problem was suppressed by the thermal control at the injector by ECAPS 52 and BUSEK of the United States.’ Citation is required for this statement.
  6. L58: ‘considered as the category micro-scale flow.’
  7. L61: ‘LBM’ should be spelled out when it is used for the first time.
  8. L64: Same for ‘DNS’. Shall spell out.
  9. L67: ‘At present, there are only a few…’
  10. L68: ‘… which limits the understanding…’
  11. L83: ‘Storage tanks include…’
  12. L132: ‘types’ instead of ‘kinds’
  13. Figure 4: Resolution is rather low, and it is difficult to distinguish the different lines. Please replace it with a higher resolution graph.
  14. L157-158: Citation is needed for ‘Lee model’
  15. L171-172: ‘…the physical properties of the propellants are assumed to be constants.’ A similar statement appears in L166-167.
  16. L194: ‘MPa’ instead of ‘mpa’
  17. L199: Spell out ‘FLIR’
  18. L208: ‘Heated to the specified temperature at the 20th hour’. Isn’t it 20 seconds?
  19. L214: ‘Single-phase flow’ instead of ‘one-way flow’
  20. Figure 5: Typo in the second figure, should be ‘Temperature’
  21. L225-226: ‘around 0.5MPa’ and ‘about 0.5MPa’
  22. L248: What is ‘anterior surface’?
  23. ‘heat is gradually trans-249 ferred upstream’. This statement appears twice in L249-250 and L252
  24. L264, L267: ‘…shows a cloud of temperature distribution…’, ‘contour’ is the more appropriate term that ‘cloud’
  25. ‘At this time’ appears for too many times in a paragraph (L281, 282, 286, 288)

The following major revisions are recommended:

  1. L93-94: It was written that ‘Capillary outlet area was heated by propane flame.’ How does the temperature controlled during the heating process? Additional description/clarification is required.
  2. L112-114: It is unclear how this statement on NTO and MMH is relevant to existing study.
  3. Figure 2: What is ‘front chamber’? This is not combustion chamber, isn’t it? Please clarify with additional description.
  4. L138: ‘…the calculated results have little change…’ How much is the change?
  5. L139-140: ‘Therefore, considering the comprehensive influencing factors of calculation efficiency and total mesh number on calculation accuracy.’ This statement does not seem to link with the previous statement. Please check and rewrite for better clarity.
  6. L175-178: It is suggested to include a screen-shot or image to show the BCs set in the simulation.
  7. L191-193: ‘When 1.5s was found through simulation calculations, …’ It is unclear what this part of statement means. Please clarify further.
  8. L198-199: ‘Use computer software Mthings and CatmanEasy to collect flow and pressure data during the experiment.’ There is no link between this statement and other parts. Please check and clarify.
  9. L216-220: The statement is unclear and fail to elaborate the results in Figure 5. A re-phrase is necessary.
  10. Figure 7: The difference in infrared images between 723, 823, 923K cases is not obvious.
  11. Figure 11: Only a qualitative analysis was given. A more detailed quantitative analysis supported by sufficient explanation is required.
  12. Figure 12: Similar to Figure 11, the analysis given is insufficient. For example, why there is a sudden increase from 0.06-0.08 s? What causes this phenomenon?  

Author Response

Thank you for your comments on this article. Please see my response comments in the attached document.

Reviewer 2 Report

The authors provide a very interesting piece of work, which combines experimental and simulation results of a thruster component, based on ADN-green propellant. 

The simulation approach seems to include the conjugate fluid-solid problem, which is quite relevant for accurate results. Nevertheless, a major revision of the presentation and organization of the results, as well as an in-depth re-wording of many expressions should be taken into account. 

Line 12 (Abstract): "... and perhaps cause phase change..": Is it not really clear if this occurs from the experiment examination? Or is it rather sensitive to variations of the injection conditions? It should be clarified this way of expressing it.

Line 60 (Introduction): "...by experiment method, and...": -> "... by an experimental method...". Improve this sentence. 

Line 64 (Introduction): "--- fluid shear forced ...": reword the sentence. It is confusing in its present state.

Line 65: "The research shows..." : -> which research? Authors' research? Please, indicate which one.

Line 67-69: the expression should be reworded.

Regarding the experimental system (from line 80, on), the language is confusing since the authors enumerate the components of the facility, including the computer; and the components include a post-processing system among them. But this one is not related to the computer; instead, it corresponds to a kind of recipient to collect the propellant products (a "tank" is depicted in Figure 1) exhausted by the thruster. The usage of the terms is far from adequate: a clear distinction between propulsion hardware itself and data adquisition-processing equipment should be done.

The authors are inaccurate at expressing several ideas, as for example in line 91: "This test uses high-pressure nitrogen as the pressure source of the propellant." It should be said: "This test uses high-pressure nitrogen to pressurize the propellant tank". A whole revision of the expression and terms should be done. Coherency across the text should be checked. For example, it is said that a "post-processing system" is part of the facility; and in Figure 1 this is referred as "Tank". The same term should be used everywhere in the text. 

The text of Figure 1 is repetitive and must be compacted; for example: "Figure 1. Schematic (a) and Image (b) of the ADN-based propellant capillary flow experimental system."

Line 92: authors write "... high-pressure nitrogen passes through the pressure reducing valve...". It is unclear (at this position in the text) if this valve sets (or not) a definite pressure value downstream its location in the piping; when this happens, the proper name of the hardware is "pressure regulator", common as part of propulsion subsystems. Please, be aware of the specific terms and the role of the facility components. In Line 194 the authors write that a "pressure of the upstream tank was stabilized at 0.5mpa,...", which seems to be the pressure level set downstream the pressure regulator. The information is written unorganized across sections of the manuscript. By the way, what is the "upstream tank"? Again, coherence in terminology must be enforced.

Text of Table 1: "Table 1. The instrument specifications" -> "Table 1. Instrument specifications"

Line 110: "3. Thruster model and grid division"-> "3. Thruster model" . I do not understand the sense of "division" in the expression "grid division": the "grid" itself is a discretization (that is, a kind of division) of the physical space where the numerical simulation is conducted. Thus, using "division" is repetitive.

Line 112: where is the acronym "NTO" explained? The same for "MMH". Anyway, this first paragraph comprising Lines 112-114 has no sense within this subsection, as it is not connected to what follows. 

Lines 116-117: The authors write "In present numerical simulation, the ADN-based thruster is simplified. The capillary structure was extracted separately". I understand that the thruster is not really simplified, but the capillary injector is the component the authors want to analyse by modeling. Again, there is a lack of accuracy in the expression. Besides, the authors should better explain why they have decided to include the front-chamber as part of the modeled component, since the decision regarding which portion of the whole thruster is focused in the simulation is -many times- driven by a simplification and/or knowledge of the boundary conditions that must be imposed. It would be fine to link this decision to the available experimental information and how this is exploited as input to the simulations. 

Line 118: the CAD software SolidWorks has been cited, so a reference must be included; the same for the ICEM software.

Line 124: "The grids were meshed using ICEM" -> It is repetitive. Better: "Grids were generated using ICEM".

Line  124-125: "Since the simplified geometry is cylindrical geometry". The capillary has a wavy form, it is not cylindrical; it is a 3D component in strict sense as represented in the FIgure. Therefore, the authors generate 3D-grids of O-topology each. I infer what they mean, but the expression in imprecise.  

The manuscript is full of this lack of accuracy at communicating the ideas and results across the whole text. So a mandatory effort to improve the manuscript should be conducted by the authors. The above mentioned examples are only some of many more found in the manuscript. A revision in depth must be done.

Major drawbacks concerning confidence and repetitivity of their findings:

Line 132: the authors say that 5 grids have been tested, and grid-independence considerations follow. Since the grids correspond to O-topology and the 5 griddings are quite progressive in size, it seems that only the 38,900 and 78,900-cell grids are different enough to be informative. Therefore, their grid-independent results, summarized in Figure 4, are not conclusive; and by no means the 55500-cell grid is a priori good choice to state that it is an adequate grid for the following two-phase flow simulations. Besides, as shown in Figure 3, cells size by the wall is too large to correctly capture the two-phase annular pattern. Really, Figure 3 presents grid-independent results for a temperature profile instead of for the sensitivity to predict a two-phase pattern inside the capillary, which is the major objective of the research.    

Line 176: Emisivity of the capillary is given but its material is not specified (which is its thermal conductivity of the solid portion of the capillary?)

In section 4, there is a sub-section (4.1) which comprises the governing equations of the two-phase flow. But the modeling -governing equations- of the solid portion of the capillary (thermal heat conduction and radiative heat emission to the ambient) is missing. There is no description regarding the conjugate fluid-solid & solid-ambient heat transfer problems, which should be better summarized in another subsection into section 4, since the inclusion of this physics is fundamental to attain accurate results in micron-scale simulations.

Author Response

(The authors gave the same response as above.)

Round 2

Reviewer 1 Report

The authors have addressed the comments accordingly and made substantial revisions to the manuscript. 

Reviewer 2 Report

The manuscript has been improved by the authors and they have answered the questions in a satisfactory manner. However, many typos can be found across the text, then a revision of English grammar should be done. Some examples:

Line 20: "...exit of capillary,..." -> "...exit of the capillary,..."

Line 35: " ...of Liquid based..." -> "...of liquid based..."

Line 36: "... and Hydrazinium nitroformate, etc." -> "... and hydrazinium nitroformate, etc."

Line 65: "... by an experiment method..." -> "...by an experimental method..."

Line72: " Mosayeb Shams [16] found..." -> "Shams et al. [16] found..."

Line 77: "... listed above [12-16], there are only a few scholars have..." -> "... a few scholars which have..."

Line 91: "...in Figure1" -> "in Figure 1."

Line 93: " ...measurement system. storage tanks..." -> "... measurement system, storage tanks..."

Line 122: " In formula d_R is..." -> "In formula (1), d_R is..."

Line 135: "... other components. the..." -> "other components.The..."

Line 140: " Therefore, in present..." -> " Therefore, in the present ..."

Line 143-145: identical to lines 140-142

Line 163: "... of grid solution, i.e... " -> "...of grids, i.e..."

Line 164: "... 78939 girds ..." -> "79839 grids..."

The manuscript continues with a high rate of typos (lines 167, 169, 170, 179, 180... )

A carefull check of the English grammar should be accomplished.